# The Impact of Environmental Habitats and Diets on the Gut Microbiota Diversity of True Bugs (Hemiptera: Heteroptera)

**DOI:** 10.3390/biology11071039

**Published:** 2022-07-11

**Authors:** Guannan Li, Jingjing Sun, Yujie Meng, Chengfeng Yang, Zhuo Chen, Yunfei Wu, Li Tian, Fan Song, Wanzhi Cai, Xue Zhang, Hu Li

**Affiliations:** 1Department of Entomology and MOA Key Lab of Pest Monitoring and Green Management, College of Plant Protection, China Agricultural University, Beijing 100193, China; liguannan@cau.edu.cn (G.L.); sunjingjing@cau.edu.cn (J.S.); cfyang07@cau.edu.cn (C.Y.); insectchen625@cau.edu.cn (Z.C.); wuyunfei@cau.edu.cn (Y.W.); ltian@cau.edu.cn (L.T.); fansong@cau.edu.cn (F.S.); caiwz@cau.edu.cn (W.C.); 2College of Food Science and Nutritional Engineering, China Agricultural University, Beijing 100193, China; mengyujie@cau.edu.cn

**Keywords:** heteroptera, gut microbiota, 16S rRNA, Reduviidae, *Enterococcus*, diet, ecological niche

## Abstract

**Simple Summary:**

There is a wide variety of insects in the suborder Heteroptera (true bugs), with various feeding habits and living habitats. Microbes that live inside insect guts play critical roles in aspects of host nutrition, physiology, and behavior. However, most studies have focused on herbivorous stink bugs of the infraorder Pentatomomorpha and the gut microbiota associated with the megadiverse heteropteran lineages, and the implications of ecological and diet variance have been less studied. Here, we investigated the gut microbial biodiversity of 30 species of true bugs representative of different ecological niches and diets. Proteobacteria and Firmicutes dominated all samples. True bugs that live in aquatic environments had a variety of bacterial taxa that were not present in their terrestrial counterparts. Carnivorous true bugs had distinct gut microbiomes compared to herbivorous species. In particular, assassin bugs of the family Reduviidae had a characteristic gut microbiota consisting mainly of *Enterococcus* and different species of Proteobacteria, implying a specific association between the gut bacteria and the host. These findings reveal that the environmental habitats and diets synergistically contributed to the diversity of the gut bacterial community of true bugs.

**Abstract:**

Insects are generally associated with gut bacterial communities that benefit the hosts with respect to diet digestion, limiting resource supplementation, pathogen defense, and ecological niche expansion. Heteroptera (true bugs) represent one of the largest and most diverse insect lineages and comprise species consuming different diets and inhabiting various ecological niches, even including underwater. However, the bacterial symbiotic associations have been characterized for those basically restricted to herbivorous stink bugs of the infraorder Pentatomomorpha. The gut microbiota associated with the megadiverse heteropteran lineages and the implications of ecological and diet variance remain largely unknown. Here, we conducted a bacterial 16S rRNA amplicon sequencing of the gut microbiota across 30 species of true bugs representative of different ecological niches and diets. It was revealed that Proteobacteria and Firmicute were the predominant bacterial phyla. Environmental habitats and diets synergistically contributed to the diversity of the gut bacterial community of true bugs. True bugs living in aquatic environments harbored multiple bacterial taxa that were not present in their terrestrial counterparts. Carnivorous true bugs possessed distinct gut microbiota compared to phytophagous species. Particularly, assassin bugs of the family Reduviidae possessed a characterized gut microbiota predominantly composed of one *Enterococcus* with different Proteobacteria, implying a specific association between the gut bacteria and host. Overall, our findings highlight the importance of the comprehensive surveillance of gut microbiota association with true bugs for understanding the molecular mechanisms underpinning insect–bacteria symbiosis.

## 1. Introduction

Insects are broadly associated with symbiotic microorganisms that enable them to consume diverse diets and occupy different ecological niches [1,2]. The insect order Hemiptera represents one of the best-studied insect lineages with respect to bacterial symbioses [3,4]. Morphologically characterized by piercing–sucking mouthparts, most hemipteran insects generally feed on a nutritionally restricted plant sap diet, e.g., phloem or xylem with high contents of carbohydrates but deficient in essential amino acids [5,6]. Three of the four hemipteran suborders, including Sternorrhyncha (psyllids, whiteflies, aphids, coccoids), Auchenorrhyncha (cicadas, spittlebugs, leafhoppers, treehoppers, planthoppers), and Coleorrhyncha (moss bugs), have engaged in an intimate relationship with maternally transmitted intracellular symbiotic bacteria, conferring on the host the ability to supplement limited nutrition resources [4,7,8,9].

With more than 42,000 described species in about 90 families and 7 infraorders, Heteroptera (true bugs) represent the largest hemipteran suborder [10]. Inhabiting various terrestrial and aquatic habitats and engaging in feeding habits that range from predation upon other arthropods and hematophagy on vertebrates to mycophagous and herbivorous habits, true bugs are also one of the ecologically most diverse and speciose lineages of insects [10,11,12]. Their diet was previously reported as one of the most prominent factors in shaping the gut microbial community [13,14,15,16]. Pentatomomorpha (>16,000 species) comprises the second greatest species-level diversity within the seven heteropteran infraorders, which is thought to be positively correlated with phytophagous feeding habits [10,17]. A plethora of studies have identified the symbiotic bacteria of phytophagous pentatomomorphans and elucidated the nutritional contributions of microbial symbionts to insect hosts [18,19,20,21]. Pentatomoidea and Coreoidea harbor monophyletic bacterial symbionts localized in specialized midgut crypts that are transmitted by postnatal transmission mechanisms [18,22,23,24]. Even specialized bacteriomes have been found in some species of Lygaeoidea [19,25,26]. However, Pyrrhocoridae have lost their crypts and harbor a stable bacterial consortium made up of two Actinobacteria (*Coriobacterium glomerans* and *Gordonibacter* sp.), one Firmicute (*Clostridium* sp.), and one Gammaproteobacterium (*Klebsiella* sp.) in the bulbus-like M3 region of their midgut [23,27]. Among the gut bacteria, *Coriobacterium glomerans* was reported to be vertically transmitted to the offspring through egg smearing, benefitting the host by vitamin supplementation [23].

Apart from Pentatomomorpha, the predatory lifestyle was maintained in the common ancestors of the other six heteropteran infraorders (Dipsocoromorpha, Enicocephalomorpha, Gerromorpha, Nepomorpha, Leptopodomorpha, and Cimicomorpha) [10,17]. Among these predator true bugs, most of the relevant studies have focused on the blood-sucking kissing bugs (triatomine Reduviidae) and bed bugs (Cimicidae) in Cimicomorpha. For example, the symbiotic bacterial associations with the genus *Rhodnius* were extensively investigated due to its involvement in Chagas disease, and symbiotic *Rhodococcus* (Actinobacteria) was identified in their gut cavity [28]; bed bugs were associated with *Wolbachia* (Alphaproteobacteria) in specialized bacteriomes [29]. Both symbionts complement the hosts with B vitamins that are limited in blood meal [29,30].

The aquatic environment was documented to contribute to the diversity of the gut microbiota of insects [16]. True bugs from Gerromorpha and Nepomorpha have successfully colonized the aquatic environment and even certain marine habitats [12], but the way in which aquatic ecology affects the gut microbiota assembly of these aquatic true bugs remains largely unknown.

Given the megadiversity of true bugs, investigations into their association with symbiotic bacteria have been restricted mainly to phytophagous insects, and lack a comprehensive characterization of the associated gut bacterial community with many major branches. In this study, we conducted a more comprehensive investigation on the gut microbiota of true bugs with different living habitats and feeding habits using 16S rRNA amplicon sequencing. Their gut bacterial profiles were investigated and compared, through which the impact of environmental habitats and diets on the gut microbiota diversity of true bugs was discussed.

## 2. Materials and Methods

### 2.1. Taxon Sampling

We collected 30 true bug species representing 8 families (Notonectidae, Belostomatidae, Nepidae, Reduviidae, Nabidae, Miridae, Aradidae, and Pyrrhocoridae) from 3 infraorders (Nepomorpha, Cimicomorpha, and Pentatomomorpha). These bugs also represent two distinct habitats (aquatic and terrestrial habitat) and three feeding habits (carnivory, herbivory, and mycophagy). The detailed collection information is provided in Appendix A. The voucher specimen of each species was kept at −80 °C at the Entomological Museum of the China Agricultural University.

### 2.2. DNA Extraction and 16S rRNA Gene Amplicon Sequencing

The gut bacterial community of selected true bugs was determined by next-generation sequencing of the V3-4 region of the 16S rRNA gene amplified from the DNA extraction of the homogenized guts of 2–15 individuals (Appendix A). The guts were dissected from adult insects using a pair of fine forceps under a dissection microscope (Nikon SMZ18, Tokyo, Japan) in a Petri dish filled with phosphate-buffered saline (PBS, 1 M). The dissected guts of the same species were pooled into one 1.5 mL microcentrifuge tube, homogenized with 1 mL of PBS, and subjected to DNA extraction using a cetyltrimethyl ammonium bromide (CTAB) based method as previously described [31]. Briefly, each sample was centrifuged, and the bacterial pellet was re-suspended in 485 µL of CTAB lysis buffer (100 mM Tris-HCl, pH 8, 1.4 M NaCl, 20 mM EDTA, 2% *w*/*v* CTAB) with 1.3 µL of β-mercaptoethanol and 13.3 µL of proteinase K (10 mg/mL). Then, the samples were transferred to bead-beating tubes with zirconia–silica beads (0.1 mm) and homogenized on a bead beater (Tissuelyser-32, Jingxin, Shanghai, China) two times for 90 s at a speed of 6.0. The samples were incubated at 56° C overnight. Then, 5 µL of RNase (10 mg/mL) was added to each sample, followed by incubation at 37 °C for 1 h. After that, the DNA was extracted with PCI (phenol–chloroform–isoamyl, 25:24:1) and precipitated with 1/10 vol NaOAc (3 M, pH 5.2) and 2.5 vol 96% ethanol, and 6 µL of glycogen (20 mg/mL) was added as a DNA carrier. The DNA pellets were re-suspended in 20 µL of H_2_O.

The bacterial 16S rRNA gene fragments (V3-V4) were amplified with primers 338F (5′-ACTCCTACGGGAGGCAGCAG-3′) and 806R (5′-GGACTACHVGGGTWTCTAAT-3′). The amplification conditions were as follows: initial denaturation for 3 min at 95 °C, followed by 27 cycles of denaturation for 15 s at 95 °C, then annealing and extension for 1 min at 55 °C. The PCR products were recovered using 2% agarose gel, and the recovered products were purified using the AxyPrep DNA Gel Extraction Kit (Axygen Biosciences, Union City, CA, USA). The recovered product was quantified with a Quantus™ Fluorometer (Promega, Madison, WI, USA). The amplicons were subjected to paired-end sequencing on the Illumina MiSeq sequencing platform (Illumina, Inc., San Diego, CA, USA) using PE300 chemical from Majorbio Bio-Pharm Technology Co., Ltd. (Shanghai, China).

### 2.3. Bioinformatic Analyses

The pair-end sequences were joined using FLASH v.1.2.11 [32] and quality-filtered with Fastp v.0.19.6 [33]. Subsequently, the joined sequences were imported to the Quantitative Insights Into Microbial Ecology (QIIME v.2.0) pipeline [34], and the DADA2 plugin [35] was used for quality filtering, denoising, chimera removal, and merging the sequences to amplicon sequence variants (ASVs). The QIIME 2 “FeatureTable [Frequency]” artifact, which contains counts of each unique sequence in each sample in the dataset, was obtained. According to the summaries of our feature table, 40 K reads on average were generated per sample, with the least number of reads being 16 K. To exclude the impact of uneven sequencing depth on the gut bacterial composition analysis, we performed alpha rarefaction to subsample the feature table at a sequencing depth of around 14 K reads per sample, and the rarefaction curve showed that the richness was saturated based on our sampling depth. Then, the ASVs were taxonomically classified using a naïve Bayes classifier pre-trained on a 16S SILVA reference (99% identity) database v.138 (QIIME2 feature-classifier plugin). This classifier was trained on the SILVA 16S rRNA database v.138 [36], where the sequences have been trimmed to only include the region that was sequenced in this analysis (the V3–V4 region, bound by the 338F/806R primer pair). The taxa plugin was used to remove all unassigned, chloroplasts, and mitochondria from the dataset. Then, an ASV table containing the relative abundance of gut bacteria in each sample was generated.

The low-abundance ASVs (<0.1% in each of the samples) were removed, and the same phylum- and genus-level ASVs were merged to generate a phylum-level table and genus-level table. The bar plot of the bacterial relative abundance at the phylum level was made by the package ggplot2 v.3.3.5 (https://rdocumentation.org/packages/ggplot2/versions/3.3.5, accessed on 10 February 2022) in R Studio v.3.6.3 (https://rstudio.com/products/rstudio/, accessed on 10 February 2022). The alpha diversity index, including Pielou’s evenness, Shannon, Simpson, and Chao, was calculated with Mothur v.1.30 [37] based on the ASVs, and the box plots were made using GraphPad Prism v.8.0.2 (GraphPad Software, San Diego, CA, USA). For β-diversity analysis, principal coordinate analysis (PCoA) based on Bray–Curtis distances was performed using the R package vegan v.2.5-7 (ANOSIM, 999 permutations; https//cran.r-project.org, accessed on 10 February 2022)) in R to describe the difference in gut microbial diversity across groups. The taxa that explained these differences were identified with linear discriminant analysis (LDA) effect size [38].

### 2.4. Determination of Gut Bacterial Number Using qPCR

DNA samples used for 16S rRNA gene amplicon sequencing were subjected to qPCR estimation of the absolute bacterial numbers. The universal 16S rRNA primers, uni1-F (5′-AGGATTAGATACCCTGGTAGTCC-3′) and uni1-R (5′-YCGTACTCCCCAGGCGG-3′) [39], were used in this study. The qPCR reactions were carried out in 25 μL of reaction mixtures on a CFX96 Real-Time PCR Detection System (Bio-Rad, CA, USA), using SYBR^®^ Premix Ex Taq™ II (TaKaRa, Kusatsu, Japan) as a fluorescent marker according to the manufacturer’s instructions. Standard curves were established by serial dilutions of plasmid DNA containing the targeted sequence as previously described [40]. The concentrations of the plasmid in these dilutions ranged from 10^2^ to 10^8^ copies per μL. Three technical replicates were employed for each sample, and negative controls were set up by replacing the template DNA with ddH_2_O to eliminate the possibility of DNA or primer dimer contamination. The gene copies of gut microbiota were calculated by comparing the Cq values (quantification cycle) to the standard curve.

## 3. Results

### 3.1. The Gut Bacterial Community Profile of Heteropteran Insects

A total of 4101 amplicon sequence variations (ASVs) were detected. Except for those unassigned and sporadically identified in a few species with a low frequency (<1% in all species), a total of nine bacterial phyla were detected to be distributed across all 30 true bug species (Appendix A). Among them, Proteobacteria (60% of the classified sequences), and Firmicute (31% of the classified sequences) were the two most predominant phyla, with high frequency in almost all of the insect species, and with phylum Proteobacteria alone accounting for over 90% in the majority of Nabidae, Miridae, and Arididae species. The bacterial phylum Actinobacteriota also presented an appropriate proportion (up to ~20%) across partial species from all eight families, while the bacterial phyla Bacteroidota, Desulfobacterota, and Patescibacteria were highly frequent in specifically aquatic species, but extremely rare in terrestrial species (Figure 1A,B and Appendix A).

### 3.2. Aquatic and Terrestrial Predatory True Bugs Had Distinct Gut Bacterial Community Profiles

As aquatic species possessed multiple specific gut bacterial phyla, we then wondered how this contributes to the fine taxonomic scale composition variance between these two ecological niches. Since the 16S rRNA gene sequence lacks the resolution of different bacterial species [46], the detected ASVs were classified to a taxonomic scale of bacterial genera. After removing those below 0.1% abundance, a total of 416 bacterial genera were detected across all species (Appendix A). To exclude the influence of dietary variance on the gut microbiota, only carnivorous insects were compared between the two ecological niches. The gut microbiota of the aquatic species differed from those of the terrestrial species. Firstly, most aquatic species, especially those of Notonectidae and Belostomatidae, harbored a gut bacterial community of high richness (Table 1) but lacked dominant bacterial taxa, with the top 30 genera detected at relatively low frequencies (average of 0.53, Figure 2A and Appendix A). This gut bacterial community profile was also seen in the related water striders (Gerridae, Gerromorpha), which also engage in an aquatic habitat lifestyle [47]. However, the diversity of the gut microbiota decreased in the species of Nepidae to a level comparable with those terrestrial carnivorous true bugs counterparts (Table 1). In contrast, almost all terrestrial predatory species were low in bacterial richness and dominated by few bacterial genera (less than four), which collectively accounted for over 90% of the abundance. Secondly, distinct gut bacterial compositions were observed between the two ecological niches, as indicated by significant segregation of the clustering in the principal coordinate analysis (PCoA) based on the Bray–Curtis distance (aquatic and terrestrial, Figure 2B, Kruskal, R = 0.41, *p* = 0.02). This is in line with the observation that multiple bacterial genera, including *Desulfovibrio*, *Phaeovibrio*, *Dysgonomonas*, *Candidatus, Soleaferrea*, and an unknown genus of the family Ruminococcaceae, were specifically pervasive in the aquatic species, with high frequencies, but extremely rare in the terrestrial counterparts (Figure 2A and Appendix A).

### 3.3. The Association between Gut Microbiota and Feeding Habits

For the terrestrial true bugs, a shift in feeding habits from carnivory to herbivory occurred in Cimicomorpha, while different feeding habits evolved in Pentatomomorpha, with Aradidae specifically engaging in a mycophagous diet [10,17]. The impact of the dietary variance on the gut microbiota of the terrestrial true bugs was accessed. It was noticed that mycophagous flat bugs (Aradidae) possessed gut microbiota of higher richness (Table 1). The PCoA clustering showed that the gut microbiota could distinguish the bugs according to their diets (Figure 2C, R = 0.32, *p* = 0.006). Particularly, carnivorous true bugs clustered separately from the other two diet groups, and herbivorous true bugs clustered together with mycophagous true bugs. We then wondered if the fine-scale host phylogeny had an impact on the gut microbiota diversity, as the gut microbiota were clustered using PCoA on the criteria of the host family (Figure 2D, Kruskal, R = 0.72, *p* = 0.001). Specifically, Pyrrhocoridae and Reduviidae clustered separately, while the other families showed overlapped clustering with each other. Particularly, Nabidae, Miridae, and Aradidae had similar gut bacterial composition profiles despite having different feeding habits, characterized by being frequently infected by the reproductive manipulator bacteria of genera *Wolbachia*, *Reckettsia*, or *Spiroplasma*. This is in line with a previous study that showed that *Wolbachia* and *Rickettsia* were pervasive in the gut of Miridae and colonized in the gut lumen as well as inside the epithelium cells [48]. Except for reproductive manipulator bacteria, different Alphaproteobacteria or Gammaproteobacteria were distributed among host species and accounted for a high proportion of the gut bacterial communities (Figure 2A and Appendix A). Interestingly, although multiple bacterial genera, e.g., *Pantoea* and *Cedecea*, shared dominant components of the gut bacterial communities of Nabidae and Miridae, the gut of Aradidae was dominated by different bacterial taxa (Figure 2A and Appendix A), suggesting that phylogenetic distance might have a potential influence on the gut microbiome assembly.

The gut microbiota of Pyrrhocoridae were characterized by a consortium of bacterial members from Actinobacteria, Proteobacteria, and Firmicutes, which is in accordance with previous findings by Sudakaran et al. [27]. Particularly, the Actinobacteria of genera *Coriobacterium* and *Gordonibacter*, which were documented as being nutritionally beneficial symbionts of *Pyrrhocoris* hosts, through the supplementation of B vitamins [20], were also frequently identified in the bugs of the family Pyrrhocoridae (Figure 2A and Appendix A). However, Actinobacteriota were extremely rare in *Dindymus rubiginosus* engaging in a predatory lifestyle, from which the two convergent bacterial genera *Coriobacterium* and *Gordonibacter* were absent (Appendix A). This might be explained by the fact that B vitamins supplied by *Coriobacterium* and *Gordonibacter* were not deficient in the carnivorous diets.

### 3.4. Assassin Bugs Had Characterized Gut Bacterial Community

The gut bacterial community of assassin bugs (Reduviidae) was distinct from those of other terrestrial families, characterized by a predominant constitution of *Enterococcus* (Firmicutes), whether or not they co-colonized with different members of Proteobacteria (Figure 2A). *Enterococcus* was the typical bacterial genus for Reduviidae (Appendix A), although its frequency differed substantially among different species (~20% to over 90%), with *Sycanus falleni* almost exclusively dominated by *Enterococcus* (Figure 2A and Appendix A). Among the species with relatively low frequencies of *Enterococcus*, two Enterobacteria genera were dominant (>20%) across different species. The bacteria genus *Yokenella* dominated in *Sycanus szechuanus* and *Pahabengkakia piliceps*, and there was a high frequency of the genus *Proteus* in *Sphedanolestes impressicollis* and *Rhynocoris fuscipes* (Figure 2A and Appendix A). *Platymeris biguttatus* was infected by parasitic bacteria of the genus *Spiroplasma*, which had a high frequency together with *Enterococcus* (Figure 2A and Appendix A).

To further explore if a specific relationship existed between assassin bugs with *Enterococcus*, the fine taxonomic scale composition of *Enterococcus* was analyzed. It was revealed that, in total, four *Enterococcus* ASVs (represented as a strain-level taxonomic resolution) were distributed across all assassin bugs, with either of the two ASVs that were predominant in one host species (Figure 3A). The ASV compositional pattern was also observed for those dominant Enterobacteria of over 20% in proportion (*Yokenella* and *Proteus*). The characteristics of a single ASV were specifically associated with one host species, despite multiple individuals being pooled together for gut microbiota determination. These ASVs were extremely rare in the other true bugs characterized in this study. By performing a BLAST search in the NCBI database, one of the *Enterococcus* ASVs dominant across *Sycanus croceovitlatus*, *Sycanus falleni*, *Sphedanolestes impressicollis*, and *Rhynocoris fuscipes* had BLAST hits to identical sequences detected from the guts of southern green stink bugs [49,50]. This result suggests that the same *Enterococcus* bacteria with an identical 16S rRNA sequence might have been convergently selected by both Reduviidae and Pentatomidae.

To check the consistency in the gut microbial compositions across individuals of the same species, the guts of four individuals of *Sycanus croceovittlatus* were dissected, and DNA was extracted for 16S rRNA amplicon sequencing (Figure 3B). The results show that the gut bacterial compositions were consistent across the individuals, despite the variance in the frequency of different gut members. *Enterococcus* was the core member in all four individuals as well as in the pooled sequencing result; however, the Proteobacteria differed in taxonomy between two sequences, as *Serratia* and *Proteus* dominated in the individual sequencing, and *Hafnia*, *Serratia*, and an unknown genus of the family Enterobacteriaceae dominated in the pool sequencing result. As functional traits are strongly conserved with their phylogenetic positions [51], it was implied that proteobacteria might be selected for their similar functions and assembled into the gut microbial community with alternations in the fine-scale taxonomy.

Functional gut microbiota generally reaches an appropriate number, such as in the gut of Pyrrhocoridae bugs [27] and the distantly related honeybee (Hymeoroptera) [52]. Given the conserved gut microbial community of Reduviidae bugs, the gut bacterial size was determined and compared across heteropteran families characterized in this study (Figure 3C). The results show that the bacterial size varied among host families as well as across species within the same family, with numbers ranging from 10^4^ to 10^10^. The gut bacterial number was relatively low in bugs of Notonectidae and Aradidae (10^4^–10^6^). However, most of the other heteropteran families accommodated a high number of gut bacteria. Among Pyrrhocoridae bugs, the gut bacterial number was approximately 10^8^ and as high as 10^9^, which was consistent with a previous determination [27]. Within Reduviidae, although the gut bacterial number varied across host species, most had a gut bacterial number of over 10^7^ and up to 10^9^ (as in *Sycanus croceovittatus* and *Sphedanolestes impressicollis*), implying that the gut microbiomes specifically assembled in Reduviidae bugs had high numbers, but not likely due to the temporary bacterial residue from diet or the environment.

## 4. Discussion

Hemiptera represent one of the best-studied lineages for their association with enormous bacterial symbionts; however, the studies have been mainly restricted to plant-feeding and blood-feeding species. Within this megadiverse insect order, a large number of species engage in different feeding habits and ecological habitats, yet there is still a lack of comprehensive investigation into the association with bacterial partners. The gut microbiota of insects, as well as other animals, provide the host with key functions in diet digestion [53,54], abiotic stress detoxification [55], nutrition supplementation [56], pathogen resistance [57,58], etc. In this work, insects of three heteropteran infraorders were selected for the characterization of their gut microbiota. The ways in which the gut microbiota assembly is affected by ecological niche and diet variance, as well as the host phylogenetic distance, were investigated.

The results show that Proteobacteria and Firmicute were the dominant phyla of all 30 true bug species. In addition, Actinobacteriota were widely distributed in all samples, although the relative abundance was low. The composition of the intestinal bacteria of aquatic stink bugs and terrestrial stink bugs was slightly different. In addition to Proteobacteria and Firmicute, aquatic stink bugs also possessed Bacteroidota, Desulfobacterota, and Patescibacteria. In line with previous surveillance, Proteobacteria, Firmicute, and Actinobacteria were consistently constituent phyla of the gut microbiota of Pyrrhocoridae [27], although Actinobacteria in one species (*Dindymus rubiginosus*) were extremely low, which is in line with the results of a previous observation [59].

When comparing the carnivorous true bug habitats in aquatic or terrestrial ecological niches, it was noticed that some species of the aquatic habitat possessed a different gut bacterial community, characterized by high bacterial richness. Similarly, water striders (Gerridae, Gerromorpha) engaging in an aquatic habitat lifestyle, such as Nepomorpha bugs, also possessed a high richness of gut bacteria but showed a distinct bacterial taxa composition [47]. Multiple aquatic true bugs were also found to be characterized by multiple specific bacterial genera compared to the terrestrial true bugs, among which, the genera *Phaeovibrio* and *Desulfovibrio* were sulfate- and nitrate-reducing bacteria and pervasive in the water environment, despite a wide distribution across a spectrum of different environments [60,61,62]. These unique bacterial components might represent a specific niche, implying the structure of the assembly of the gut bacterial community. These results suggest that the gut bacterial community of aquatic predatory true bugs differs in both diversity and composition from those of terrestrial true bugs.

Diet was previously reported as one of the most prominent factors in shaping gut microbial community [13,15,16]. Consistently, the diet is also the criterion that could partially distinguish the gut bacterial community [13]. However, mycophagous true bugs could not be clustered separately from herbivorous true bugs, which might be attributed to the lower number of mycophagous species collected. More broad taxon sampling is required for further characterizing the difference in the diversity of the gut microbiota between these two feeding habits. The gut microbiota of Pyrrhocoridae were characterized by a consortium of bacterial members from Actinobacteria, Proteobacteria, and Firmicutes, which is in accordance with the previous findings of Sudakaran et al. [27]. It was reported previously that *Pyrrhocoris apterus* and *Dysdercus fasciatus* harbored a gut bacterial community constituting *Coriobacterium glomerans* and *Gordonibacter* sp. of Actinobacteria, *Clostridium* sp. of Firmicutes, and *Klebsiella* sp. of Gammaproteobacteria [63]. Particularly, the Actinobacteria of genera *Coriobacterium* and *Gordonibacter* were documented to be nutritionally beneficial symbionts of *Pyrrhocoris* hosts through the supplementation of B vitamins [20]. Consistently, true bugs from the same genus in our study also possessed the same two bacterial genera, while the bacteria from Proteobacteria and Firmicutes varied at lower phylogenetic scales at the genus level. This result suggests that except for the hosts of nutritionally beneficial Actinobacteria, other bacterial taxa might not be specifically associated with the host. However, Actinobacteriota were extremely rare in *Dindymus rubiginosus* engaging in a predatory lifestyle, from which the two convergent bacterial genera *Coriobacterium* and *Gordonibacter* were absent (Appendix A). This might be explained by the fact that B vitamins supplied by *Coriobacterium* and *Gordonibacter* were not deficient in a carnivorous diet.

It was revealed that assassin bugs (Reduviidae) possessed a characteristic gut microbiota, with the same *Enterococcus* strains that consistently dominated the gut bacterial community across different species. *Enterococcus* was also detected as a dominant gut bacterial member in blood-feeding kissing bugs *Rhodnius prolixus* and *Triatoma vitticeps* [64], suggesting that it might be a conserved gut component to most assassin bugs despite their diverse feeding habits. One *Enterococcus* strain was also previously detected in some species of stink bugs (Pentatomomorpha) [49,50], which might imply convergent gut selection by the hosts, or less likely, ancestrally inheritance. Even though no inflated gut structure was visualized from the external gut morphology, internal changes (e.g., crypt structures extended to the gut lumen) could not be excluded for the accommodation of the large number of gut bacteria. Further investigation, including quantifying the *Enterococcus* distribution along gut sections, FISH visualization, axenic host construction, and nutrition evaluations, will be necessary for a comprehensive understanding of the interactions between the host and gut *Enterococcus*.

The specific association of *Enterococcus* with assassin bugs indicates that a potential interaction between them might have evolved. Normally, predatory insects are considered to be independent of nutritional symbionts, but many of them need defensive bacterial partners. As an example, *Steinernema* nematodes require the assistance of symbiotic *Xenorhabdus* bacteria for prey digestion and to suppress the proliferation of other microorganisms that compete for resources from dead prey [65,66]. A recent study also showed that a carnivorous diet was preferential for Enterobacteria, including some entomopathogenic bacteria in the gut of plant bugs, compared to an herbivorous diet, which was detrimental to bugs’ survival [67]. *Enterococcus* in stink bugs and other herbivorous insects was reported to be implicated with diet digestion and the detoxification of plant defensive chemicals [68,69,70]. *Enterococcus* dominated in the gut of generalist herbivore cotton leafworm (*Spodoptera littoralis*) and was documented to secrete bacteriocin against invading bacteria, providing a defensive function to the host [71]. More empirical tests are needed to validate whether *Enterococcus* in the gut of assassin bugs plays a similar defensive role for its hosts.

This work represents a comprehensive survey of the gut microbiota of the megadiverse heteropteran insects, showing that varied ecological niches affected the assembly of gut microbiota and that phylogeny plays a dominant role in shaping the gut bacterial community over the effects of dietary variance. Assassin bugs might have an intimate relationship with gut *Enterococcus*. The bacterial symbiotic associations with true bugs need more intensive studies, with respect to the ecological, economical, and biological implications.

## 5. Conclusions

Hemiptera represent one of the best-studied lineages for their association with enormous bacterial symbionts; however, the studies have been mainly restricted to plant-feeding and blood-feeding species. Within this megadiverse insect order, large numbers of species engage in different feeding and ecological habitats, yet there is still a lack of comprehensive investigation into their association with bacteria. Based on a broader taxon sampling of true bugs, the current study shows that varied ecological niches affected the assembly of gut microbiota, and both the diets and host phylogeny participated in shaping the gut bacterial community over the effects of dietary variance. Assassin bugs might have an intimate relationship with gut *Enterococcus*. The bacterial symbiotic associations with true bugs need more intensive studies, with respect to the ecological, economical, and biological implications.

## Figures and Tables

**Figure 1 biology-11-01039-f001:**
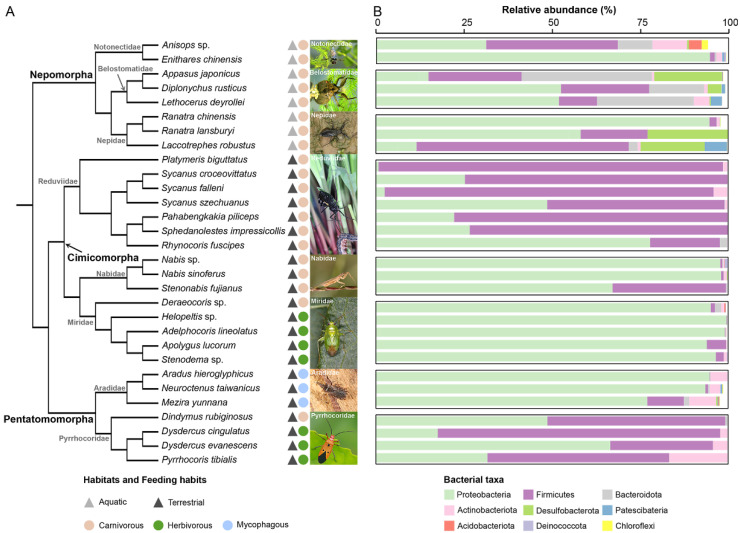
The gut bacterial composition at the phylum level for 30 heteropteran species. (**A**) Phylogeny of the host species. The phylogenetic tree of the 30 selected heteropteran species was reconstructed using Mesquite 3.51 [41], with the topology constrained to recently published phylogenetic studies of true bugs [42,43,44,45]. Insect infraorder and family are indicated at nodes. The habitat ecology and feeding habit are labeled by color. Typical ecological photos of true bugs from the same families are also shown and all photos are taken by authors. (**B**) The relative abundance of gut bacterial phylum estimated from the 16S rRNA gene amplicon sequencing data. Bacterial phyla with abundance of below 1% are not shown.

**Figure 2 biology-11-01039-f002:**
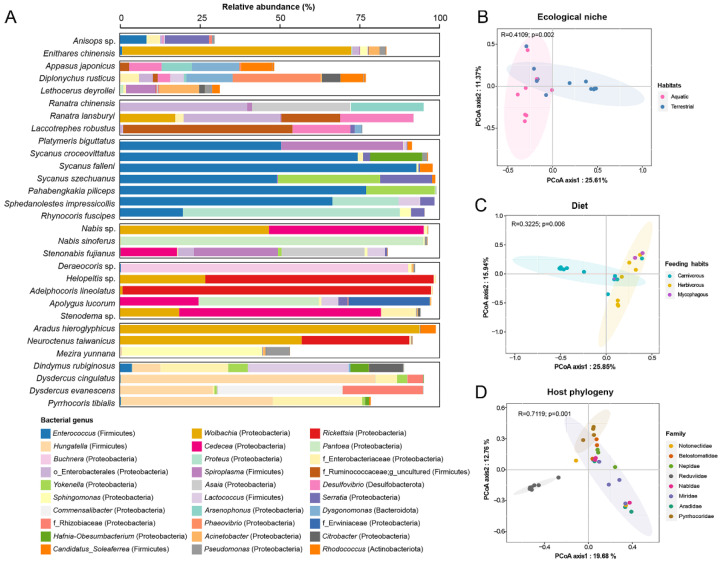
The gut bacterial community profiles differed between environmental habitats and across host taxonomy. (**A**) The frequency of the top 30 most abundant bacterial genera across different species. For the reads that were not able to be assigned to the taxonomic scale of genera, the family name prefixed by the letter “f” is given. (**B**–**D**) The PCoA clustering based on the Bray–Curtis distances of the microbiota of carnivorous true bugs between aquatic and terrestrial habitats (**B**), terrestrial true bugs of different diets (**C**), or different families (**D**).

**Figure 3 biology-11-01039-f003:**
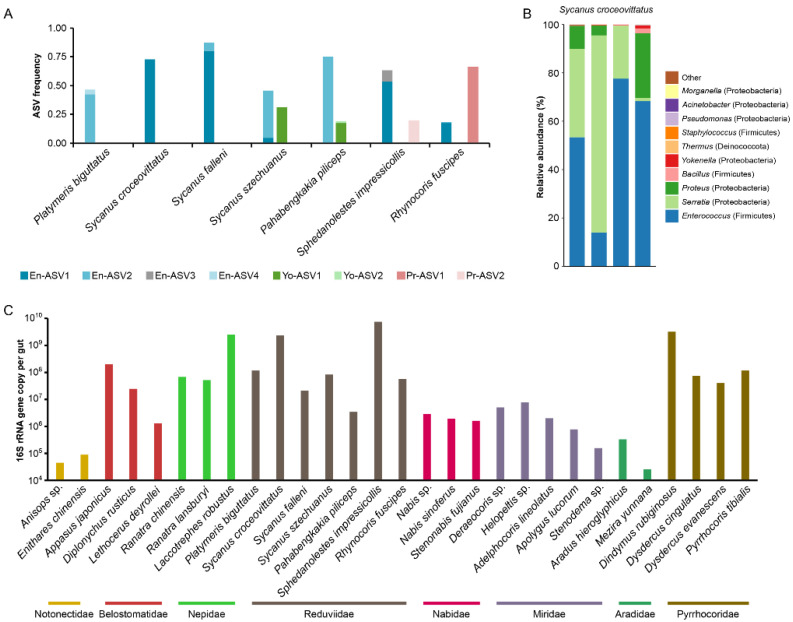
(**A**) The ASV compositions corresponding to the dominant bacterial genera (>20%) comprising the gut microbiota of assassin bugs. The ASVs from the same bacterial genus are stacked together. (**B**) The frequencies of the top 10 abundant bacterial genera across individuals of species *Sycanus croceovittatus*. (**C**) The quantitative results for the gut bacterial size of bugs among different heteropteran families.

**Table 1 biology-11-01039-t001:** Richness and diversity of the gut bacterial community.

Host Taxon		Host Species	Pielou’s Evenness	Chao1	Shannon	Simpson’s Index of Diversity
Infraorder	Family					
Nepomorpha	Notonectidae	*Anisops* sp.	0.847	203.00	4.22	24.39
		*Enithares chinensis*	0.344	90.50	1.55	1.93
	Belostomatidae	*Appasus japonicus*	0.662	93.09	2.79	11.63
		*Diplonychus rusticus*	0.635	117.50	2.65	8.20
		*Lethocerus deyrollei*	0.805	158.67	3.62	20.00
	Nepidae	*Ranatra chinensis*	0.353	36.43	1.41	3.29
		*Ranatra lansburyi*	0.634	19.00	1.70	4.74
		*Laccotrephes robustus*	0.527	55.75	1.75	3.09
Cimicomorpha	Reduviidae	*Platymeris biguttatus*	0.367	32.00	1.12	2.47
		*Sycanus croceovittatus*	0.386	9.00	0.87	1.72
		*Sycanus falleni*	0.261	10.00	0.36	1.16
		*Sycanus szechuanus*	0.482	17.00	1.15	2.70
		*Pahabengkakia piliceps*	0.210	26.60	0.63	1.56
		*Sphedanolestes impressicollis*	0.606	5.00	0.99	2.03
		*Rhynocoris fuscipes*	0.465	25.33	1.09	1.99
	Nabidae	*Nabis* sp.	0.240	75.00	1.02	2.22
		*Nabis sinoferus*	0.374	56.38	0.38	1.12
		*Stenonabis fujianus*	0.514	48.33	1.91	5.18
	Miridae	*Deraeocoris* sp.	0.223	53.20	0.68	1.24
		*Helopeltis* sp.	0.232	37.25	0.72	1.72
		*Adelphocoris lineolatus*	0.066	42.50	0.25	1.08
		*Apolygus lucorum*	0.603	40.00	1.58	3.70
		*Stenodema* sp.	0.555	27.00	1.24	2.25
Pentatomomorpha	Aradidae	*Aradus hieroglyphicus*	0.151	27.00	0.31	1.14
		*Neuroctenus taiwanicus*	0.309	120.46	1.28	2.29
		*Mezira yunnana*	0.598	134.91	2.68	4.67
	Pyrrhocoridae	*Dindymus rubiginosus*	0.642	35.67	2.13	5.78
		*Dysdercus cingulatus*	0.392	28.60	0.89	1.56
		*Dysdercus evanescens*	0.472	29.25	1.41	3.34
		*Pyrrhocoris tibialis*	0.547	53.88	1.49	3.13

## Data Availability

The original 16S rRNA sequence data are available in the NCBI Sequence Read Archive under accession number PRJNA804667.

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
