# Peer review of "The Impact of Environmental Habitats and Diets on the Gut Microbiota Diversity of True Bugs (Hemiptera: Heteroptera)"

_biology, 2022, doi:10.3390/biology11071039_

Round 1

Reviewer 1 Report

Dear Authors,

this is a nice manuscript and i have not much to criticize. 

please, find below a few comments for improvement of this manuscript.

- Table 1. the interpretation of the Simpson index as used here is difficult as its values are counterintuitive, small value = high diversity. Rather use the inverse Simpson index (1/Simpson), that give more meaningful values. Along with the diversity indices you should also calculate the eveness, as this will describe how much skew you have in the distribution. Furthermore, you should discuss Diplonychus rusticus as it shows a mjor differences between observed and estimated (Chao) taxa, reasons for this could be....?

- for several analyses you could use much better phylogenetic correction methods, e.g. HyPhy or similar ones that finally give you phylogenetically independent contrasts. this will be helpful for measures like number of taxa, diversity of microbiome etc. If highly related taxa show a signal, that implies there might be a single origin only and sampling several taxa within such a group might amplify this signal, although it belongs to a rather unique and not general class. The phylogenetic correction that you use by just taking families into consideration is inappropriate, as we don't know whether the diversity and taxon representation within families are comparable to each other and we don't know how distant (phylogenetically) all the families are from each other. 

Author Response

Responses to Reviewer #1:

Dear Authors,

This is a nice manuscript and I have not much to criticize.

Please, find below a few comments for improvement of this manuscript.

Reply: We appreciate the reviewer’s positive evaluation of our work. 

1. Table 1. the interpretation of the Simpson index as used here is difficult as its values are counterintuitive, small value = high diversity. Rather use the inverse Simpson index (1/Simpson), that give more meaningful values. Along with the diversity indices you should also calculate the eveness, as this will describe how much skew you have in the distribution.

Reply: As recommended by the reviewer, Table 1 was modified with Simpson’s Index of Diversity (1-Simpson‘s index (D)) was calculated and replace the Simpson’s index, and the Pielou’s evenness was also added.

2. Furthermore, you should discuss Diplonychus rusticusas it shows a mjor differences between observed and estimated (Chao) taxa, reasons for this could be....?

Reply: Thanks for the reviewer’s comment. The major differences between observed and estimated (Chao) taxa was due to that different taxonomic scale was used. As the Chao estimated the number of taxa at 99% 16S identity as referred to the SILVA reference database v.138. We showed here the genus level taxa number which was used as the classification unit and discussed in the following manuscript. The discrepancy between Chao and observed genera number implying that multiple taxa with 99% 16S rRNA sequence identity might be identified within genus Diplonychus rusticus. To avoid further confusing for authors, we also removed the column of observed genera number in Table 1, as it provided not much new information comparing Chao1 index.

3. For several analyses you could use much better phylogenetic correction methods, e.g. HyPhy or similar ones that finally give you phylogenetically independent contrasts. this will be helpful for measures like number of taxa, diversity of microbiome etc. If highly related taxa show a signal, that implies there might be a single origin only and sampling several taxa within such a group might amplify this signal, although it belongs to a rather unique and not general class. The phylogenetic correction that you use by just taking families into consideration is inappropriate, as we don't know whether the diversity and taxon representation within families are comparable to each other and we don't know how distant (phylogenetically) all the families are from each other.

Reply: We appreciate the reviewer’s constructive suggestion for the use of phylogenetic correction methods. We have studied the software very carefully and find that this kind of method is very helpful for measure the phylogenetic signal of biological characters and their relationships. However, a comprehensive species sampling along host insect phylogeny is recommended for this analysis. Given the taxon sampling in the present study is still limited , we will apply this method in the future study based on a more comprehensive phylogenetic framework with more intensive sampling from seven infraorders of the suborder Heteroptera. In fact, we are already working on this extremely challenging work.

Reviewer 2 Report

General comment: The manuscript provides novel information regarding the gut microbiota across 30 species of true bugs representative for 36 different ecological niches and diets. The authors have compared the impact of environmental habitats and diets on the gut microbiota of true bugs. The manuscript is well written, organized, and clear, with exception of some clarifications pointed below. I recommend acceptance for publication after minor changes.

Key words

Line 47: I would change ‘16S’ to ‘16S rRNA'

Introduction

Line 102: ‘Their gut bacterial profiles were …’

Materials and Methods

Did authors do a control for potential contaminations by sequencing negative controls and mock communities?

Lines 141-143: ‘To explore the taxonomic composition…’ Please rewrite this sentence.

2.2. DNA extraction and 16S rRNA gene amplicon sequencing: I would briefly explain how the library preparation was done.

Results and Discussion

3.2. Aquatic and terrestrial predatory true bugs had distinct gut bacterial community profiles: As indicated in Table S1, the bugs have been collected from various sites. Please explain how and in what extent collection sites can affect the results. Why did the authors not analyze environmental samples to see the overlap of microbes between the gut of the bugs and the environment?

Author Response

Responses to Reviewer #2:

General comment: The manuscript provides novel information regarding the gut microbiota across 30 species of true bugs representative for 36 different ecological niches and diets. The authors have compared the impact of environmental habitats and diets on the gut microbiota of true bugs. The manuscript is well written, organized, and clear, with exception of some clarifications pointed below. I recommend acceptance for publication after minor changes.

Reply: We appreciate the reviewer’s positive evaluation of our work. 

Key words

Line 47: I would change ‘16S’ to ‘16S rRNA'

Reply: Thanks for pointing this out, we have modified this in the new text.

Line47: Keywords: Heteroptera; gut microbiota; 16S rRNA; Reduviidae; Enterococcus; diet; ecological niche

Introduction

Line 102: ‘Their gut bacterial profiles were …’

Reply: The new text was revised accordingly.

Line 102: Their gut bacterial profiles were investigated and compared, through which, the impact of environmental habitats and diets on the gut microbiota diversity of true bugs was discussed.

Materials and Methods

Did authors do a control for potential contaminations by sequencing negative controls and mock communities?

Reply: We did not sequence negative controls. But when performing a PCR amplification of 16S rRNA gene, we did a negative control in each test (replacing the DNA template with sterile water). Then agarose gel electrophoresis detection was performed and no band was observed for the negative controls. And for each bug dissection, the gut was rinsed with PBS several times before DNA extraction.

 Lines 141-143: ‘To explore the taxonomic composition…’ Please rewrite this sentence.

Reply: This sentence was rephrased in the new text.

Line 141-143: Then ASVs were taxonomically classified using a Naïve-Bayes classifier pre-trained on 16S SILVA reference (99% identity) database v.138 (QIIME2 feature-classifier plugin).

2.2. DNA extraction and 16S rRNA gene amplicon sequencing: I would briefly explain how the library preparation was done.

Reply: Thanks for the comment. We added a brief description on the library preparation process in the method section in the new text.

Line 122-131: Briefly, each sample was centrifuged and the bacterial pellet was re-suspended in 485 µl CTAB lysis buffer (100 mM Tris-HCl, pH 8, 1.4 M NaCl, 20 mM EDTA, 2% w/v CTAB) with 1.3 µl β-mercaptoethanol and 13.3 µl proteinase K (10 mg/ml). Then the samples were transferred to bead-beating tubes with zirconia/silica beads (0.1 mm), and homogenized on a bead-beater for two times 90 s, at speed 6.0. Samples were incubated at 56° C over-night. 5 µl RNase (10 mg/ml) was added to each sample, followed by incubation at 37 °C for 1 h. After that, the DNA was extracted with PCI (phenol-chloroform-isoamyl, 25:24:1) and precipitated with 1/10 vol NaOAc (3 M, pH 5.2) and 2.5 vol 96% Ethanol, with 6 µl glycogen (20 mg/ml) added as a DNA carrier. DNA pellets were re-suspended in 20 µl H2O.

Line 136-138: The PCR products were recovered using 2% agarose gel, and the recovered products were purified using AxyPrep DNA Gel Extraction Kit (Axygen Biosciences, Union City, CA, USA). The recovered product was quantified with a Quantus™ Fluorometer (Promega, USA).

Results and Discussion

3.2. Aquatic and terrestrial predatory true bugs had distinct gut bacterial community profiles: As indicated in Table S1, the bugs have been collected from various sites. Please explain how and in what extent collection sites can affect the results. Why did the authors not analyze environmental samples to see the overlap of microbes between the gut of the bugs and the environment?

Reply: Thanks for the comment. Yes, different collection site might have varied environmental bacterial compositions which might impact the composition of insect gut microbiome. But as there are enormous diversity of bacteria among environments, particularly in soil and aquatic environments where the bugs characterized in this study lives, such that during the lifecycle of the bugs, they could have opportunities to contact with much diverse bacterial taxa. The environmental samples collection and sequencing could give more comprehensive infomations, e.g. the the overlapped microbes between the gut of the bugs and the environments. However, for each bug species, multiple environmental specimens were required. Given the big number of true bug species characterized here, this would substantially increase the labor and cost. Given that we couldn’t compare the environmental bacterial composition with that of the guts of true bugs, and can be conducted in further studies.